# Antifungal Effect of Plant Extracts on the Growth of the Cereal Pathogen *Fusarium* spp.—An In Vitro Study

Weronika Kursa [1], Agnieszka Jamiołkowska [1,*], Jakub Wyrostek [2] and Radosław Kowalski [2]

[1] Department of Plant Protection, University of Life Sciences in Lublin, 20-950 Lublin, Poland
[2] Department of Analysis and Food Quality Assessment, University of Life Sciences in Lublin, 20-950 Lublin, Poland
*  Correspondence: aguto@wp.pl

**Abstract:** The aim of the study was a laboratory evaluation of the antifungal effect of leaf extracts from yarrow (*Achillea millefolium* L.), tansy (*Tanacetum vulgare* L.), sage (*Salvia officinalis* L.) and wormwood (*Artemisia absinthium* L.) on fungi of the genus *Fusarium*, major cereal pathogens. The study used 5%, 10%, and 20% concentrations of plant extracts, evaluating their effect on the linear growth of *Fusarium avenaceum*, *F. culmorum*, *F. graminearum*, *F. sporotrichioides* and the percentage of their growth inhibition compared to control. The study also included the assessment of the content of selected biologically active compounds in plant extracts and their impact on the development of the aforementioned pathogenic fungi. The total content of polyphenols and flavonoids in the extracts was assessed by spectrophotometry, and antioxidant activity was determined using the synthetic 2,2-diphenyl-1-picrylhydrazyl (DPPH) radical. Plant extracts from sage were characterized by the highest polyphenol contents (81.95 mg/mL) and flavonoids (21.12 mg/mL) compared to other plant extracts, and also showed the highest antioxidant activity (102.44 mM Trolox). Wormwood extract contained the lowest amount of phenolic compounds (flavonoids—5.30 mg/mL, polyphenols—43.83 mg/mL). Plant extracts inhibited the mycelia growth of fungal pathogen depending upon the fungus species, type of extract and its concentration. The extracts of sage (S) and tansy (T) plants at a concentration of 20% demonstrated strong inhibitory effect against the tested fungi (the highest inhibition coefficient for S20: 83.53%; T20: 72.58%), while 10% and 5% extracts of these plants were less effective in inhibiting the growth of *Fusarium* (highest inhibition coefficient for S10: 71.33%; S5: 54.14%; T10: 56.67%; T5: 38.64%). Yarrow (Y) and wormwood (W) extracts showed low fungistatic effect. Their 20% concentration inhibited the development of mycelia growth of fungi at the level of 63.82% (W20) and 67.57% (Y20). The 5% and 10% concentrations of these plant extracts had the weakest effect on the tested fungi (Y5: 34.09; W5: 42.06%; Y10: 45.01%; W10: 57.44%), even stimulating the *Fusarium* growth compared to the control (Y5: −23.7%). Based on the study, it was found that each species of fungus reacted differently to the addition of the extract to the culture medium and its concentration, however *F. avenaceum* and *F. culmorum* were the most sensitive fungi, while the least sensitive was *F. graminearum*. The results of the research are the preliminary phase for further field tests to determine the fungistatic effect of plant extracts in field conditions, their phytotoxicity and biological stability, as well as the possibility of producing a biopreparation to protect plants against fusariosis.

**Keywords:** plant extracts; *Fusarium*; antifungal activity

## 1. Introduction

Crop plants are constantly attacked by pathogens both during pre- and post-harvest stages, often causing economically important yield losses. The main culprits of plant diseases are pathogenic fungi, which cause reduced yields, and their lower nutritional and organoleptic value [1]. In some cases, fungi are also indirectly responsible for the occurrence of allergic disorders and poisoning among consumers. Fungi of the genus *Fusarium* deserve special attention in this regard [2]. They infect cereal plants cultivated in different climatic zones [3–7], mainly causing head blight [8]. Grain is therefore the main source of inoculum

of fungi causing cereal diseases during the growing season [7,9,10]. *Fusarium* spp. can survive in the soil in the form of saprotrophic mycelium, and some species also in the form of chlamydospores [8,10]. Fungi of the genus *Fusarium* are the cause of pre-emergence and post-emergence blight of cereal seedlings. From a toxicological and economic point of view, however, the most dangerous disease caused by *Fusarium* spp. is head blight (FHB), which is accompanied by contamination of grain with mycotoxins (fumonisins, trichothecenes, and zearalenone) [4,11,12]. The ability of these fungi to produce mycotoxins is a very important factor determining the harmfulness of *Fusarium* spp. and reducing cereal grain quality [2,13]. In most regions of the world, the main causes of head blight are *F. graminearum* Schwabe, *F. culmorum* (Wm.G. Sm.) Sacc., *F. avenaceum* (Fr.) Sacc., *F. graminearum* Schwabe and *F. sporotrichioides* Sherb. [14–16]. The risk of contamination of agricultural products with mycotoxins is not limited to raw materials. Consumption of contaminated feed by livestock can lead to contamination of meat, milk, eggs and related products [17,18].

In this context, the use of synthetic fungicides is still the most effective method of protecting cereals against pathogenic fungi. However, their application causes long-term persistence of active ingredients of pesticides in food and the environment [19]. Researchers are searching for new solutions to safely protect crops against pathogens. One such effort is the search for new biologically active compounds, whose purpose is to limit the development of pathogenic fungi, while inhibiting the production of mycotoxins and having a low negative impact on the environment [20,21]. Plants are the source of many biologically active compounds, especially herbal plants, including thyme, oregano, garlic, sage [22,23]. The plants' tremendous biosynthetic capacities make it possible to use them to produce biological preparations and apply them as alternatives to synthetic chemicals [22,23]. Natural bioactive compounds called natural fungicides are non-specific, and their effect on pathogens is comprehensive [24]. The effect of natural fungicides based on plant extracts depends mainly on the content of phenols, terpenes and alkaloids [25]. Phenols have anti-radical and antioxidant properties, and selected groups of phenolic compounds, such as phenolic acids, flavonoids and tannins have an additional direct anti-fungal effect [26–28]. The mechanisms of action of these compounds are still poorly understood [28,29]. It is assumed that the antifungal properties of phenolics are attributed to their lipophilicity and/or the occurrence of the hydroxyl groups in their structure. Due to their binding properties to adhesions and proteins, they are qualified to disrupt membranes, inactivate enzymes and complex metal ions, thereby exhibiting toxic effects upon fungi. In particular, the lipophilicity of phenolics facilitates penetration of the cytoplasmic membrane, whereas hydroxyl groups are involved in the uncoupling of oxidative phosphorylation [25]. The fungistatic effect is also exhibited by alkaloids, which are usually found in poisonous plants. These compounds are effective against bacteria, parasites and fungi [28]. They penetrate the cell wall and/or DNA of the fungus [25]. Natural bioactive compounds used in plant protection not only limit the growth of fungi (fungistatic effect) but also engage plant defense responses [20,30]. Direct action of these compounds is based on inhibition of fungal sporulation, germination of spores, and reduction of hyphae growth [31]. Natural secondary metabolites produced by plants under the influence of elicitors also have a protective role in relation to pathogens [30].

The aim of the study was a laboratory evaluation of the fungistatic effect of plant extracts from tansy, yarrow, common sage and wormwood on selected *Fusarium* spp., important in agricultural phytopathology. One of the research stages was the assessment of the content of selected biologically active compounds in alcohol plant extracts and the analysis of their effect on the growth of polyphagic pathogenic fungi. The fungistatic activity of the extracts was evaluated based on the analysis of the fungal growth inhibition coefficient. The results of the conducted research are fundamental in deciding on further work addressed to the development of commercial preparations and their formulation in order to effectively protect cereals against fusarioses.

## 2. Materials and Methods

### 2.1. Chemical Characterization of Plant Extracts

#### 2.1.1. Plant Material

The research plant material consisted leaves of yarrow (*Achillea millefolium* L.), tansy (*Tanacetum vulgare* L.), sage (*Salvia officinalis* L.) and wormwood (*Artemisia absinthium* L.) collected prior to flowering of plants. The material was collected in the natural environment (Podlasie Province, Poland), dried and powdered. Raw materials were stored in sealed packages, protected from light, moisture and the influence of foreign odors. The plants used in the research are plants commonly found in the natural state of a temperate climate and are easy to obtain herbal material.

#### 2.1.2. Extract Preparation

To prepare the extracts, 300 g of each powdered herb was suspended in 3000 mL of 70%. Extraction was carried out under a reflux condenser at the boiling point of ethanol for 6 h. The resulting extract was filtered through filter paper and concentrated to 300 mL (extract 1:1) using a rotary evaporator (Heidolph Instruments, Schwabach, Germany). The test extracts were stored in a refrigerator (4 °C).

#### 2.1.3. Total Polyphenol Analysis

The concentration of total polyphenols and total flavonoids was determined in the extracts. The concentration of polyphenols (calculated as gallic acid) was determined using the spectrophotometric method ($\lambda$ = 765 nm) with the Folin–Ciocalteau reagent, according to the modified method of Singelton and Rossi [32]. The results were calculated from the equation of the calibration curve prepared for gallic acid standards in the concentration range of 10–60 mg/L (10, 20, 30, 40, 50, 60 mg/L). Each sample, depending on the initial concentration, was diluted according to the range on the standard curve. All analyses were performed in triplicate.

#### 2.1.4. Flavonoid Analysis

The content of flavonoids (in terms of epicatechin) was determined by spectrophotometry according to a modified method described by Karadeniz et al. [33]. The results were calculated based on the calibration curve prepared for epicatechin standards in the concentration range of 10–400 mg/L (10, 50, 100, 150, 200, 250, 300 and 400 mg/L). Each sample, depending on the starting concentration, was diluted according to the range of the standard curve. All analyses were performed in triplicate.

#### 2.1.5. Assessment of Extract Antioxidant Activities

The antioxidant activity of plant extracts was determined using the modified method of Brand–Willams et al. [34] using the synthetic radical DPPH (1,1-diphenyl-2-picrylhydrazyl, Sigma) converted to mM Trolox [35]. The ability of the tested antioxidant to counteract the oxidation reaction was calculated from the following formula: % inhibition = 100 $(A_0 - A_m)/A_0$, where $A_m$ is the mean absorbance of the test solution containing the antioxidant, and $A_0$ is the absorbance of the DPPH radical solution.

### 2.2. Biological Assay

#### 2.2.1. Isolation of Fungal Cultures

Fungal cultures were obtained in 2020–2021 as a result of mycological analyses of winter wheat grains (*Triticum aestivum* L.) of the cultivars 'Hondia' (DANKO Plant Breeding Company, Poland), 'Euforia' (Plant Breeding Company in Strzelce, Poland), 'Linus' (RAGT Semences Group, France). The following species were studied: *Fusarium avenaceum* (Fr.) Sacc. (strain P27), *F. culmorum* (Wm.G. Sm.) Sacc. (strain Fc37), *F. graminearum* Schwabe (strain Fg54), *F. sporotrichioides* Sherb. (strain P41). The fungal inoculum was derived from 10-day-old single-spore colonies grown on Potato Dextrose Agar (PDA Difco, Becton, Dickinson & C., France) stored in the fungal collection of the Department of Plant Protection,

University of Life Sciences in Lublin. Confirmation of the affiliation of the fungal strain to the species was made on the basis of microscopic analysis of each isolate/strain (structure and size of spores, colony colour) using appropriate mycological keys.

### 2.2.2. In Vitro Evaluation of Antifungal Potential of Plant Extracts

The study evaluated the effect of 5%, 10%, and 20% extracts of yarrow (Y), tansy (T), sage (S), and wormwood (W) on the linear growth of the fungi tested. For this purpose, the method of poisoned media was used [31]. The control consisted of fungal colonies growing on PDA medium with the addition of 5%, 10%, and 20% residue after evaporation of the extraction solvent (70% ethanol; the total volume of 1000 mL was evaporated to 100 mL in a rotary evaporator under the same conditions as for the preparation of plant extracts). The experiment was performed in 5 replicates. Experimental combinations were incubated for 10 days at 25 °C. The diameter of fungal colonies (mm) was measured after 2, 4, 6, 8 and 10 days. The measure of the antifungal effect was the inhibition of mycelial growth on the culture medium with extract addition compared to control. The fungistatic activity of plant extracts was calculated on the basis of the growth inhibition/stimulation coefficient of fungal colonies calculated using the Abbott formula: $I = [(C - T)/C] \times 100\%$, where: I—inhibition index of fungus linear growth (%), C—diameter of fungus colony in the control sample, E—diameter of fungus colony in the experimental sample containing the test substance in agar [31].

### 2.3. Statistical Analysis

Data were analyzed by analysis of variance (Duncan's test) at the $p \leq 0.05$ significance level using the Statistica program 12.6 (StatSoft Polska, Kraków, Poland).

## 3. Results

### 3.1. Content of Polyphenols, Flavonoids and Antioxidant Activity of Extracts

The chemical composition of yarrow, tansy, common sage and wormwood extracts was determined in the study. The average contents of flavonoids in the extracts are presented in Table 1. The concentration of flavonoids in the tested extracts ranged from 7.82 to 21.12 mg/mL and differed significantly ($p \leq 0.05$) depending on the type of extract. The highest content of flavonoids was recorded in sage extracts (21.12 mg/mL), and the lowest in wormwood extracts (5.30 mg/mL). The highest content of polyphenols was also found in the extracts from sage (81.95 mg/mL) and tansy (77.12 mg/mL), while significantly ($p \leq 0.05$) lower in the wormwood extract (43.83 mg/mL). The lowest antioxidant activity was recorded for yarrow extract (84.40 mM Trolox) and the highest for sage (102.44 mM Trolox) (Table 1).

**Table 1.** Concentration of flavonoids (epicatechin equivalent mg/mL), polyphenols (gallic acid equivalent mg/mL) and antioxidant activity in the basal extract (100%).

| Plant Extract | Flavonoids (mg/mL) ± SD | Polyphenols (mg/mL) ± SD | Antioxidant Activity, Free Radical-Scavenging Ability | |
|---|---|---|---|---|
| | | | % Inhibition ± SD | mM Trolox ± SD |
| yarrow (Y) | 5.05 [c] ± 0.130 | 59.56 [b] ± 4.080 | 56.13 [a] ± 0.386 | 84.40 [a] ± 0.608 |
| tansy (T) | 7.82 [b] ± 0.069 | 77.12 [a] ± 3.075 | 57.79 [a] ± 6.383 | 87.02 [a] ± 10.065 |
| sage (S) | 21.12 [a] ± 0.904 | 81.95 [a] ± 8.117 | 67.58 [a] ± 2.855 | 102.44 [a] ± 4.502 |
| wormwood (W) | 5.30 [c] ± 0.218 | 43.83 [c] ± 3.280 | 62.92 [a] ± 5.857 | 95.10 [a] ± 9.235 |

SD—Standard Deviation; a, b, c—values in the rows marked with the same letter do not differ significantly at a significance level of $p \leq 0.05$.

### 3.2. Inhibition of Fungal Growth

Laboratory tests of biotic activity allowed to determine the direct effect of plant extracts on the growth dynamics of the fungi. Each fungal species reacted differently to the addition of plant extracts in the substrate, their concentration and time of exposure. The strongest antifungal effect was recorded for extracts from sage and tansy (Figures 1 and 2, Tables 2–5).

The strength of the fungistatic effect of plant extracts increased with increasing concentrations. Extracts at a concentration of 20% showed the strongest effect.

**Table 2.** Inhibition (%) of the growth of *Fusarium avenaceum* (mm) after application of plant extracts.

| Experimental Combination | | Number of Days ± SD | | | | |
|---|---|---|---|---|---|---|
| | | 2nd | 4th | 6th | 8th | 10th |
| Y | 5% concentration | 34.09 ef ± 3.94 | 16.43 e ± 10.68 | 7.96 e ± 8.33 | −9.43 e ± 6.38 | −23.7 e ± 3.33 |
| T | | 38.64 de ± 13.64 | 23.19 e ± 12.88 | 11.07 e ± 8.45 | −10.03 e ± 8.42 | −19.4 e ± 3.03 |
| S | | 29.55 ef ± 14.19 | 54.11 bc ± 14.22 | 41.87 e ± 9.51 | 27.05 c ± 9.86 | 13.6 cd ± 10.49 |
| W | | 19.32 f ± 5.21 | 19.32 e ± 7.29 | 9.34 e ± 9.07 | −9.73 e ± 7.76 | −21.6 e ± 6.84 |
| Y | 10% concentration | 42.55 cde ± 12.77 | 20.93 e ± 13.32 | 10.83 e ± 9.38 | 8.26 d ± 7.34 | 2.77 d ± 6.43 |
| T | | 51.06 bcd ± 3.69 | 29.07 de ± 2.01 | 17.5 e ± 2.50 | 15.29 cd ± 1.06 | 7.56 d ± 3.05 |
| S | | 35.11 de ± 14.39 | 40.12 cd ± 13.09 | 31.25 d ± 8.20 | 22.94 c ± 8.75 | 19.65 c ± 7.87 |
| W | | 38.30 de ± 3.69 | 20.35 e ± 5.04 | 10.42 e ± 3.15 | 9.48 d ± 2.65 | 3.78 d ± 4.43 |
| Y | 20% concentration | 67.57 a ± 0.00 | 59.35 ab ± 3.71 | 50.3 bc ± 10.80 | 47.05 b ± 8.74 | 36.25 b ± 7.24 |
| T | | 67.57 a ± 5.41 | 65.42 ab ± 3.53 | 56.81 b ± 7.17 | 53.61 b ± 6.20 | 44.81 b ± 6.53 |
| S | | 63.96 ab ± 3.12 | 73.83 a ± 1.62 | 76.33 a ± 1.02 | 72.43 a ± 3.99 | 61.30 a ± 8.90 |
| W | | 56.76 abc ± 5.41 | 52.80 bc ± 3.53 | 51.19 bc ± 1.54 | 47.26 b ± 1.00 | 39.51 b ±0.61 |

Y—yarrow; T—tansy; S—sage, W—wormwood extract; SD—Standard Deviation; a, b, c . . . —values in the rows marked with the same letter do not differ significantly at a significance level of $p \leq 0.05$.

**Table 3.** Inhibition (%) of the growth of *Fusarium culmorum* (mm) after application of plant extracts.

| Experimental Combination | | Number of Days ± SD | | | | |
|---|---|---|---|---|---|---|
| | | 2nd | 4th | 6th | 8th | 10th |
| Y | 5% concentration | 26.98 ef ± 2.75 | 2.14 f ± 18.33 | 10.2 h ± 9.33 | 16.48 d ± 5.56 | 6.48 e ± 2.85 |
| T | | 34.92 de ± 15.31 | 11.54 f ± 23.54 | 27.86 g ± 15.30 | 28.52 c ± 14.92 | 16.30 d ± 11.98 |
| S | | 57.14 b ± 4.76 | 46.58 cde ± 1.96 | 47.01 def ± 2.69 | 45.19 b ± 1.95 | 29.07 c ± 4.72 |
| W | | 42.06 cd ± 4.96 | 35.47 e ± 7.06 | 38.31 fg ± 2.83 | 38.52 bc ± 2.10 | 23.33 cd ± 1.11 |
| Y | 10% concentration | 15.46 f ± 3.57 | 42.67 de ± 1.53 | 45.61 ef ± 2.50 | 30.93 c ± 3.16 | 16.67 d ± 2.22 |
| T | | 50.52 bc ± 12.37 | 59.67 bcd ± 13.50 | 50.76 cde ± 5.15 | 37.96 bc ± 10.28 | 23.89 cd ± 7.78 |
| S | | 54.64 bc ± 9.45 | 71.33 ab ± 4.16 | 67.56 ab ± 7.45 | 45.00 b ± 5.47 | 26.85 c ± 5.89 |
| W | | 24.74 ef ± 9.94 | 52.33 b–e ± 6.35 | 57.44 bcd ± 2.82 | 46.11 b ± 5.64 | 30.74 c ± 3.78 |
| Y | 20% concentration | 48.39 bcd ± 7.39 | 59.41 bcd ± 4.67 | 56.53 b–e ± 1.80 | 38.89 bc ± 5.47 | 24.07 cd ± 3.90 |
| T | | 72.58 a ± 2.79 | 71.47 ab ± 6.85 | 67.54 ab ± 4.58 | 56.48 a ± 2.85 | 41.85 b ± 4.98 |
| S | | 74.19 a ± 2.79 | 83.53 a ± 1.02 | 75.00 a ± 1.71 | 65.37 a ± 2.74 | 52.59 a ± 1.95 |
| W | | 61.29 ab ± 4.84 | 63.82 bc ± 6.18 | 59.7 bc ± 5.92 | 44.63 b ± 5.04 | 28.89 c ± 4.19 |

Y—yarrow; T—tansy; S—sage, W—wormwood extract; SD—Standard Deviation; a, b, c . . . —values in the rows marked with the same letter do not differ significantly at a significance level of $p \leq 0.05$.

**Table 4.** Inhibition (%) of the growth of *Fusarium graminearum* (mm) after application of plant extracts.

| Experimental Combination | | Number of Days ± SD | | | | |
|---|---|---|---|---|---|---|
| | | 2nd | 4th | 6th | 8th | 10th |
| Y | 5% concentration | 28.57 a ± 0.00 | 13.01 g ± 15.42 | 8.05 d ± 4.09 | −9.09 c ± 5.21 | −22.41 c ± 3.33 |
| T | | 28.57 a ± 0.00 | 36.99 b–e ± 4.75 | 22.88 bcd ± 31.66 | 4.17 bc ± 44.25 | −2.41 bc ± 56.23 |
| S | | 16.67 abc ± 4.12 | 23.97 efg ± 5.44 | 22.03 bcd ± 7.77 | 4.17 bc ± 11.50 | −20.00 c ± 23.29 |
| W | | 21.40 ab ± 7.14 | 21.23 fg ± 5.17 | 11.00 cd ± 5.83 | −9.84 c ± 4.59 | −27.93 c ± 6.57 |
| Y | 10% concentration | 16.67 abc ± 8.33 | 41.24 a–d ± 3.09 | 33.10 ab ± 6.37 | 30.52 ab ± 6.54 | 22.32 ab ± 7.09 |
| T | | 4.17 c ± 12.50 | 53.61 a ± 0.00 | 46.13 a ± 1.06 | 39.78 a ± 2.06 | 35.94 a ± 1.02 |
| S | | 16.67 abc ± 8.33 | 42.27 a–d ± 12.50 | 42.96 ab ± 9.21 | 41.96 a ± 10.04 | 38.39 a ± 10.46 |
| W | | 11.08 bc ± 4.81 | 51.03 ab ± 3.89 | 48.93 a ± 11.20 | 43.87 a ± 13.32 | 35.27 a ± 21.29 |

**Table 4.** *Cont.*

| Experimental Combination | | Number of Days ± SD | | | | |
|---|---|---|---|---|---|---|
| | | **2nd** | **4th** | **6th** | **8th** | **10th** |
| Y | 20% concentration | 25.71 ᵃᵇ ± 9.90 | 32.00 ᵈᵉ ± 4.00 | 37.75 ᵃᵇ ± 1.84 | 30.60 ᵃᵇ ± 0.55 | 29.95 ᵃ ± 2.74 |
| T | | 28.57 ᵃ ± 4.95 | 48.00 ᵃᵇᶜ ± 5.29 | 49.40 ᵃ ± 3.19 | 42.90 ᵃ ± 4.47 | 39.06 ᵃ ± 2.07 |
| S | | 22.86 ᵃᵇ ± 13.09 | 54.00 ᵃ ± 7.21 | 42.17 ᵃᵇ ± 3.19 | 39.75 ᵃ ± 5.78 | 36.98 ᵃ ± 3.16 |
| W | | 14.31 ᵇᶜ ± 4.95 | 34.68 ᶜ⁻ᶠ ± 13.32 | 30.12 ᵃᵇᶜ ± 4.34 | 19.55 ᵃᵇ ± 5.27 | 17.96 ᵃᵇ ± 1.35 |

Y—yarrow; T—tansy; S—sage, W—wormwood extract; SD—Standard Deviation; a, b, c . . . —values in the rows marked with the same letter do not differ significantly at a significance level of $p \leq 0.05$.

**Table 5.** Inhibition (%) of the growth of *Fusarium sporotrichioides* (mm) after application of plant extracts.

| Experimental Combination | | Number of Days ± SD | | | | |
|---|---|---|---|---|---|---|
| | | **2nd** | **4th** | **6th** | **8th** | **10th** |
| Y | 5% concentration | 3.64 ᵉ ± 15.75 | 13.67 ᶠ ± 6.01 | 21.09 ᶠ ± 2.73 | 9.63 ᵈ ± 3.57 | 0.93 ᵈ ± 1.60 |
| T | | 23.64 ᶜᵈᵉ ± 11.89 | 24.82 ᵈᵉᶠ ± 3.47 | 28.18 ᵉᶠ ± 0.36 | 27.59 ᵇᶜ ± 16.52 | 5.37 ᶜᵈ ± 4.85 |
| S | | 37.27 ᵇᶜᵈ ± 4.72 | 37.41 ᶜᵈᵉ ± 9.22 | 41.96 ᶜᵈ ± 7.12 | 25.93 ᵇᶜ ± 6.51 | 6.11 ᶜᵈ ± 3.89 |
| W | | 25.45 ᶜᵈ ± 7.87 | 32.73 ᶜᵈᵉ ± 7.19 | 37.58 ᵈᵉ ± 5.06 | 26.85 ᵇᶜ ± 5.04 | 7.96 ᶜᵈ ± 2.80 |
| Y | 10% concentration | 2.63 ᵉ ± 12.06 | 23.47 ᵉᶠ ± 3.68 | 29.01 ᵉᶠ ± 2.88 | 14.44 ᶜᵈ ± 5.36 | 4.44 ᶜᵈ ± 2.94 |
| T | | 31.58 ᶜᵈ ± 15.79 | 42.86 ᵇᶜ ± 10.75 | 43.41 ᶜᵈ ± 5.80 | 29.81 ᵇᶜ ± 5.34 | 12.96 ᵇᶜ ± 3.21 |
| S | | 43.86 ᵃᵇᶜ ± 6.08 | 46.94 ᵃᵇᶜ ± 6.12 | 48.68 ᵇᶜ ± 4.40 | 30.93 ᵃᵇᶜ ± 4.10 | 7.04 ᶜᵈ ± 5.16 |
| W | | 17.54 ᵈᵉ ± 20.44 | 38.44 ᶜᵈ ± 12.17 | 44.02 ᶜᵈ ± 9.09 | 29.26 ᵇᶜ ± 13.22 | 13.52 ᵇᶜ ± 8.36 |
| Y | 20% concentration | 60.28 ᵃ ± 2.46 | 55.21 ᵃᵇ ± 6.44 | 46.36 ᶜᵈ ± 9.68 | 35.37 ᵃᵇ ± 10.40 | 20.19 ᵃᵇ ± 13.38 |
| T | | 64.54 ᵃ ± 6.50 | 54.57 ᵃᵇ ± 13.65 | 49.80 ᵃᵇᶜ ± 6.14 | 29.26 ᵇᶜ ± 12.35 | 20.00 ᵃᵇ ± 4.34 |
| S | | 63.12 ᵃ ± 8.86 | 59.31 ᵃ ± 8.09 | 58.10 ᵃᵇ ± 2.65 | 39.07 ᵃᵇ ± 6.24 | 14.26 ᵇᶜ ± 6.86 |
| W | | 56.03 ᵃᵇ ± 2.46 | 58.99 ᵃ ± 3.94 | 58.50 ᵃ ± 3.56 | 46.11 ᵃ ± 2.94 | 26.11 ᵃ ± 5.47 |

Y—yarrow; T—tansy; S—sage, W—wormwood extract; SD—Standard Deviation; a, b, c . . . —values in the rows marked with the same letter do not differ significantly at a significance level of $p \leq 0.05$.

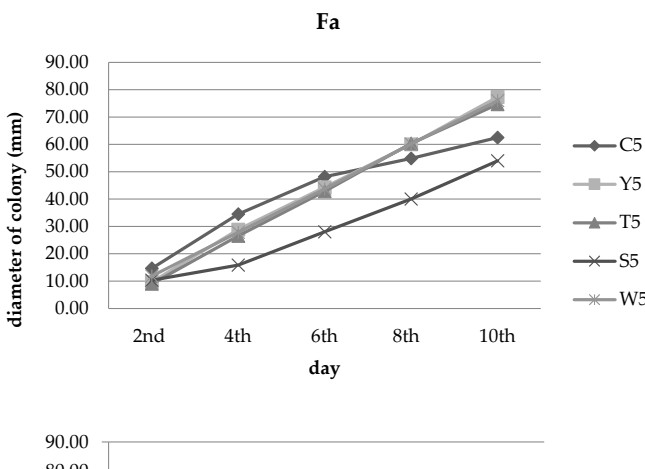

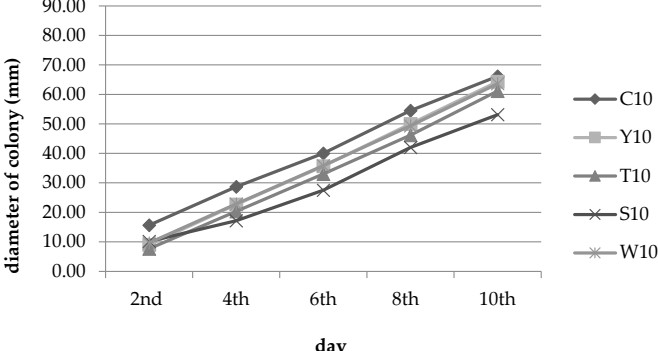

**Figure 1.** *Cont.*

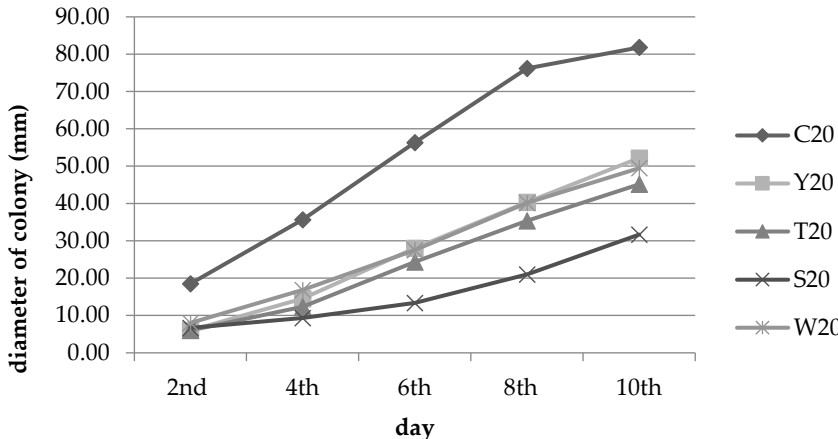

**Fc**

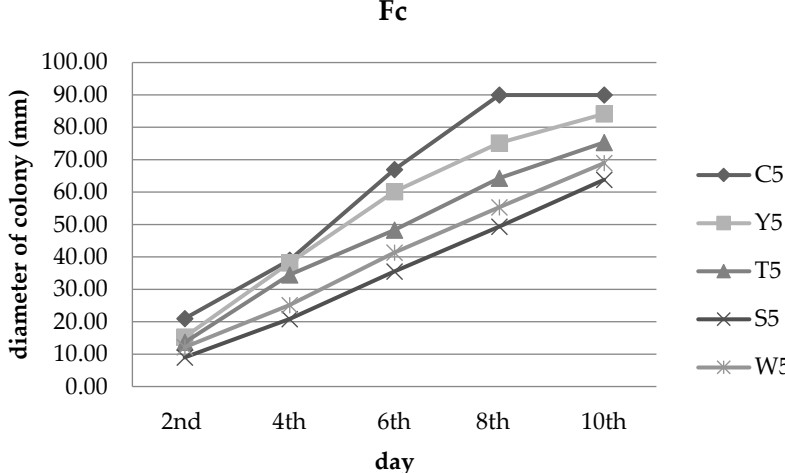

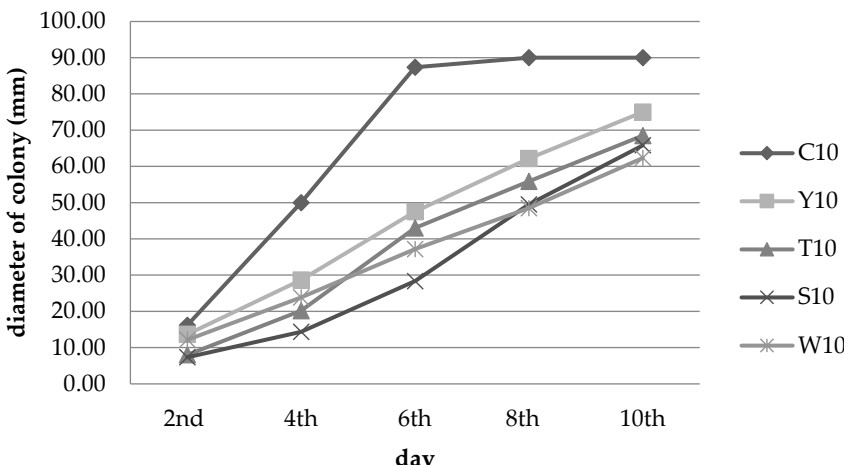

**Figure 1.** *Cont.*

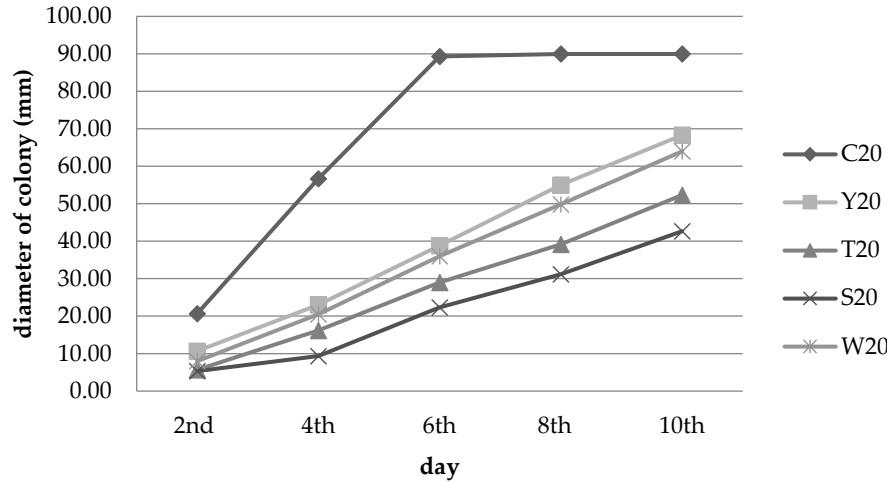

**Fg**

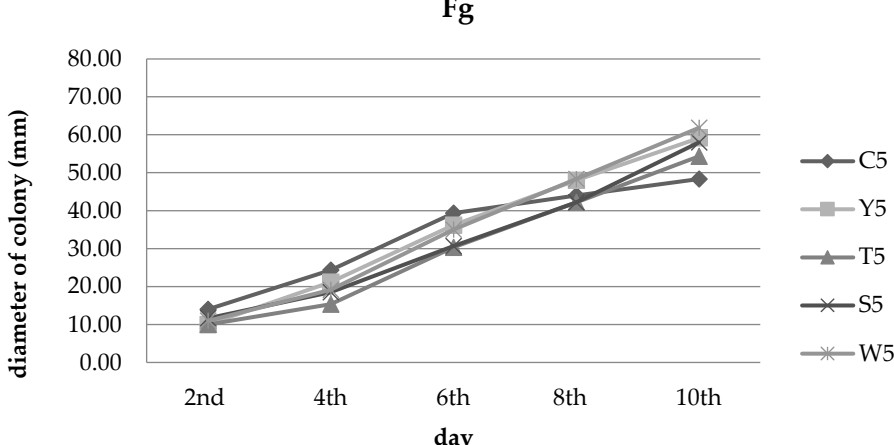

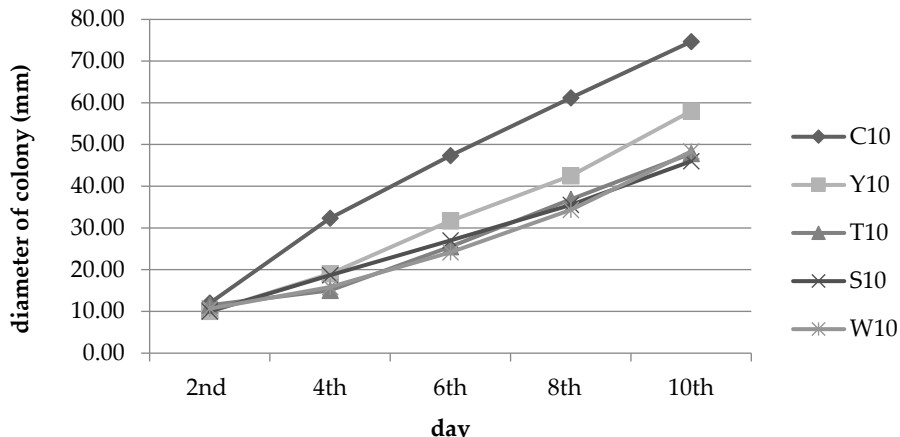

**Figure 1.** *Cont*.

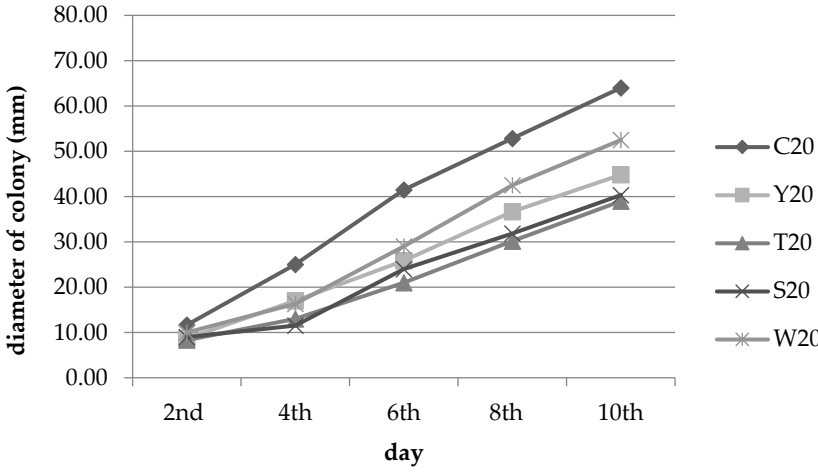

**Fs**

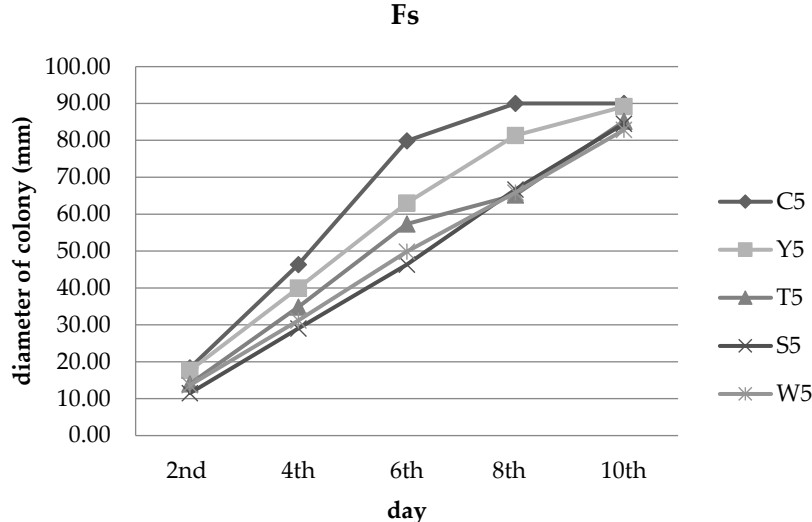

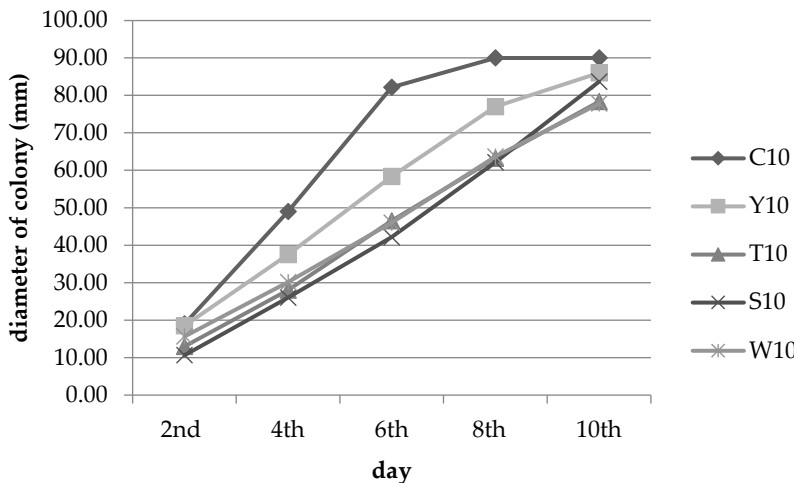

**Figure 1.** *Cont.*

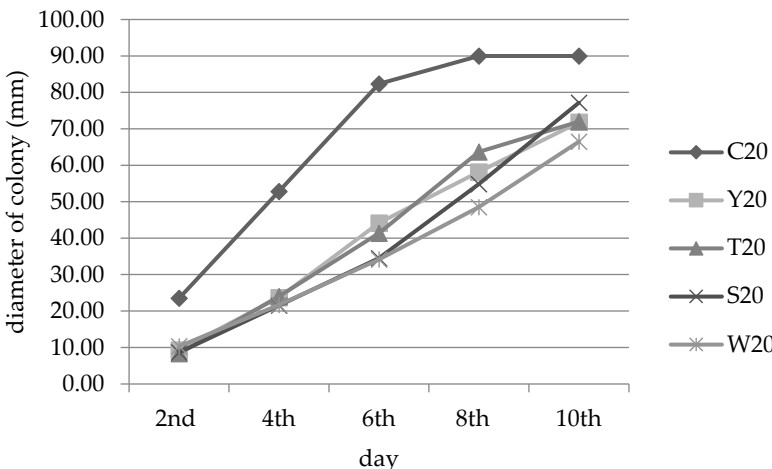

**Figure 1.** Diameter of colony (mm) of fungi growing on potato-dextrose agar (PDA) with different plant extract concentrations; Fa—*F. avenaceum*, Fc—*F. culmorum*, Fg—*F. graminearum*, Fs—*F. sporotrichioides*; C5, C10, C20—control with 5%, 10%, 20% solvent residue; Y5, Y10, Y20—5%, 10%, 20% yarrow extract concentration, T5, T10, T20—5%, 10%, 20% tansy extract concentration; S5, S10, S20—5%, 10%, 20% sage extract concentration; W5, W10, W20—5%, 10%, 20% wormwood extract concentration.

Sage extracts (S) significantly ($p \leq 0.05$) inhibited fungal growth for the duration of the experiment, and the best results were obtained for 20% extract. The best antifungal effects were recorded against *F. culmorum* (S20: 52.59–83.53%), *F. avenaceum* (S20: 61.30–76.33%) and *F. sporotrichioides* (S20: 14.26–63.12%) throughout the experiment (Figure 1, Tables 2, 3 and 5). Of the fungi tested, *F. graminearum* was the least sensitive to the 20% concentration of this extract in the culture medium (S20: 22.86–54.00%) (Figure 1, Table 4). The weakest effect of the sage extract was recorded for the concentration of 5%, especially against *F. graminearum* (S5: 4.17–23.97%), where the fungistatic effect of the extract decreased already on day 4 of the experiment, and a faster experimental colony growth was observed on days 8 and 10 day compared to control (Figure 1, Table 4). However, the antifungal effect of sage extract against *F. graminearum* was not significantly ($p \leq 0.05$) different compared to other plant extracts (Tables 2–5).

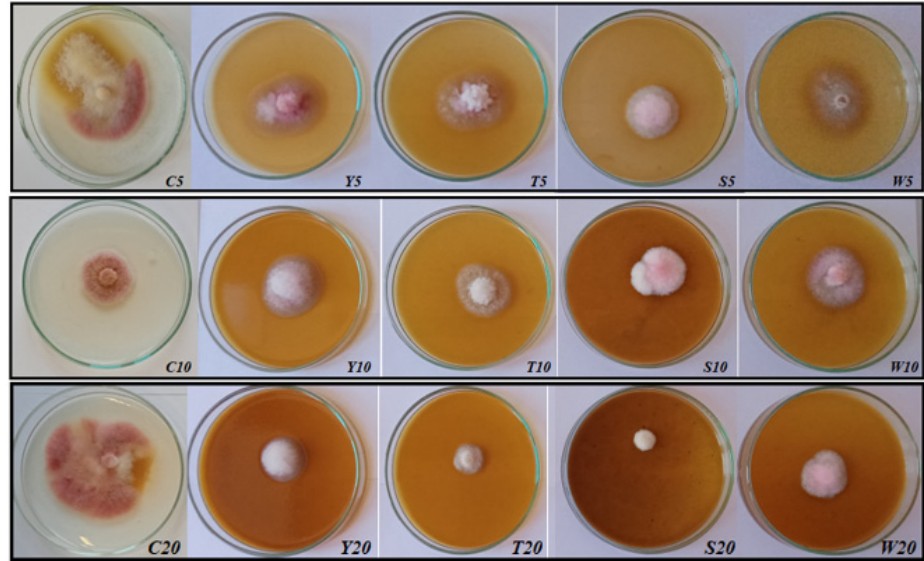

*Fusarium avenaceum*

**Figure 2.** *Cont.*

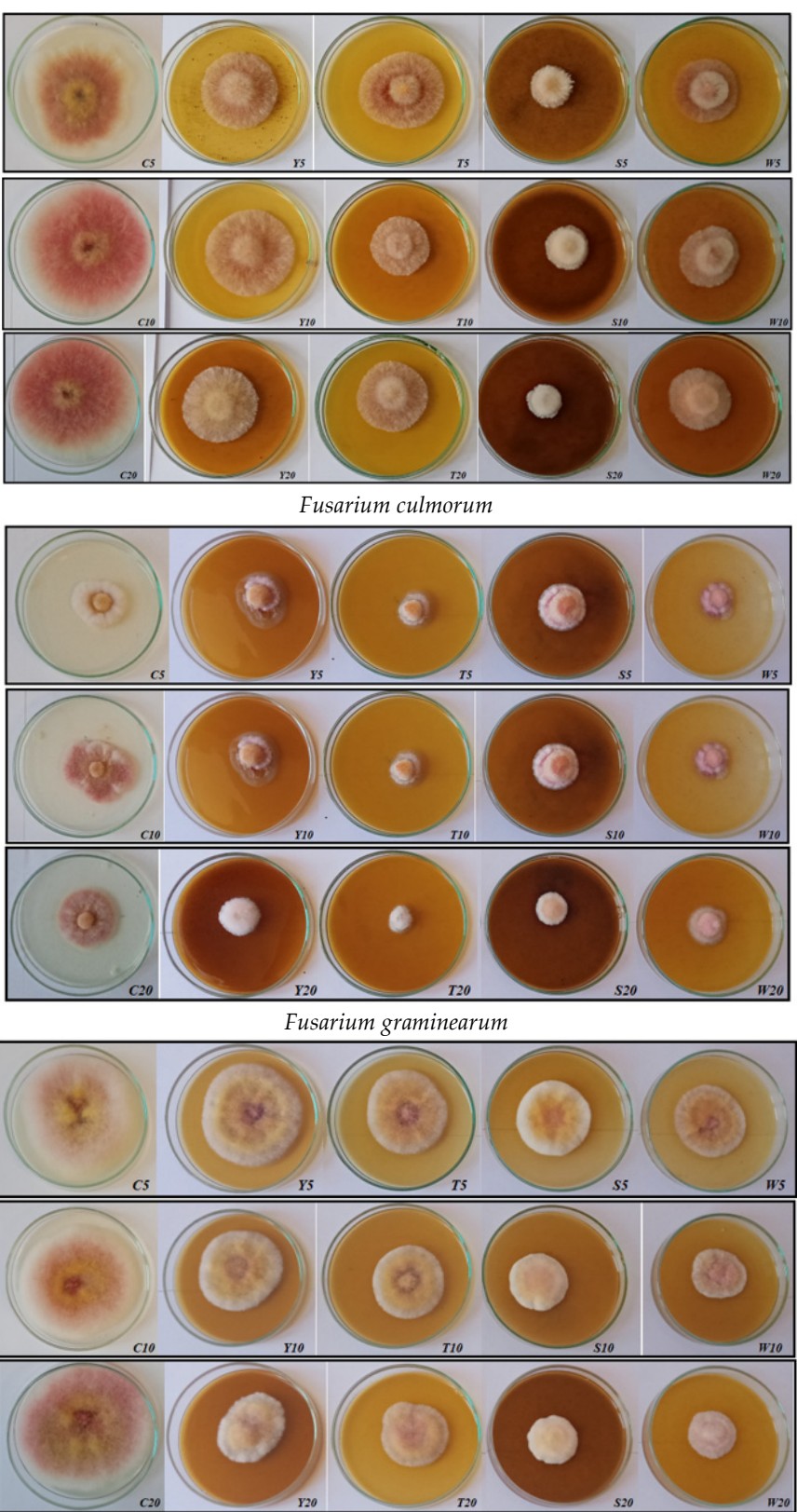

*Fusarium culmorum*

*Fusarium graminearum*

*Fusarium sporotrichioides*

**Figure 2.** Linear growth of fungal colonies on PDA with the addition of plant extracts; C5, C10, C20—control with 5%, 10%, 20% solvent residue; Y5, Y10, Y20—5%, 10%, 20% yarrow extract, T5, T10, T20—5%, 10%, 20% tansy extract; S5, S10, S20—5%, 10%, 20% sage extract; W5, W10, W20—5%, 10%, 20% wormwood extract; 6th day of growth.

Interesting results were also noted after applying 20% concentration of tansy extract (T). The fungi *F. culmorum* (T20: 41.85–72.58%), *F. avenaceum* (T20: 44.81–67.57%), and *F. sporotrichioides* (T20: 20.00–64.54%) were particularly sensitive to the effect of tansy, but the fungistatic effect of sage was not significantly ($p \leq 0.05$) different compared to the other plant extracts at this concentration for individual fungal species (exception: *F. culmorum* Y20) (Tables 2–5). Tansy extract exerted the strongest effect in the first days of the experiment (days 2, 4), while its effectiveness decreased in the following days. The weakest effect of tansy extract was recorded for a concentration of 5% and against *F. sporotrichioides* (T5: 5.37–28.18%) (Figure 1, Table 5).

Yarrow (Y) and wormwood (W) extracts showed slightly lower fungistatic activity, the weakest at a concentration of 5%, which slightly inhibited the surface growth of *F. sporotrichioides* (Y5: 0.93–21.09%; W5: 7.96–37.58%), and they even stimulated colony growth of *F. graminearum* and *F. avenaceum* on day 10 of the experiment (Tables 2, 4 and 5). The 20% concentration of yarrow extract was most effective in limiting the growth of *F. sporotrichioides* (Y20: 20.19–60.28%), and least effective against *F. graminearum* (Y20: 25.71–37.75%). Interesting results were also recorded for wormwood extracts, especially with regard to *F. culmorum* colonies, whose surface growth at the beginning of the experiment (2–4 days) was inhibited at the level of 61.29–63.82%. *F. graminearum* strain turned out to be particularly resistant to wormwood extract, and its 20% concentration inhibited the growth of the fungus only at the level of 14.31–34.68% (Tables 3 and 4). The study found that each fungal species reacted differently to the addition of the extract to the culture medium and its concentration, however, *F. avenaceum* and *F. culmorum* were the most sensitive fungi, while the least sensitive was *F. graminearum*.

The conducted experiment also focused on changes in the morphology of the fungi under the influence of plant extracts (Figure 2, Table 6). The most common were changes in mycelial structure, and coloration of the obverse and reverse of the colony. With increasing extract concentration in the medium, the changes in the colony structure intensified. Fungal colonies with plant extracts in the medium were more compact and of elevated growth, while in the control mycelium the mycelium was looser and less compact. The reverse side of the test fungi was lighter or colorless compared to the control colonies (Table 6). The fungistatic effect of plant extracts demonstrated in the study on *Fusarium* spp., i.e., species important in agricultural phytopathology, warrants further research in this area. There is a need to conduct field experiments in this area, which should evaluate the effects of individual extracts on phytotoxicity and plant health during vegetation, as well as to test their effectiveness for grain treatment.

**Table 6.** Selected features of fungal morphology under the influence of plant extracts (6th day of the experiment).

| Fungus Species | Experimental Combination | Mycelium Surface and Structure | Obverse | Reverse |
|---|---|---|---|---|
| *F. avenaceum* | C5, C10, C20 | fluffy, slightly elevated | pink-white; pink | maroon |
| | Y5, Y10, Y20 | substrate, slightly compact, elevated in the center | white-purple | purple, light purple |
| | T5, T10, T20 | slightly compact; elevated | white | slightly pink; colorless |
| | S5, S10, S20 | fluffy, slightly elevated | pink-white; white-cream | slightly pink; colorless |
| | W5, W10, W20 | flat, centrally slightly elevated | white-gray-pink | pink-white |
| *F. culmorum* | C5, C10, C20 | fluffy, regular, even growth | white-pink | purple |
| | Y5, Y10, Y20 | fluffy, slightly elevated in the center | purple | maroon |
| | T5, T10, T20 | fluffy, slightly raised | purple; pink-white | purple; maroon |
| | S5, S10, S20 | compact; elevated | white-pink | light pink |
| | W5, W10, W20 | fluffy, slightly elevated in the center | gray-pink-white | maroon |
| *F. graminearum* | C5, C10, C20 | regular growth, elevated in the center | white-pink; pink | maroon |
| | Y5, Y10, Y20 | irregular, substrate, elevated in the center | pink-purple-white | brown and maroon |
| | T5, T10, T20 | regular growth, elevated | white-pink | pink |
| | S5, S10, S20 | regular growth, elevated | white-pink-yellow | light brown |
| | W5, W10, W20 | regular growth, elevated in the center | white-purple | gray-purple; light pink |

**Table 6.** *Cont.*

| Fungus Species | Experimental Combination | Mycelium Surface and Structure | Obverse | Reverse |
|---|---|---|---|---|
| *F. sporotrichioides* | C5, C10, C20 | fluffy, regular, even growth | pink-white-yellow | maroon |
| | Y5, Y10, Y20 | fluffy, slightly raised | pink-white-yellow | maroon |
| | T5, T10, T20 | fluffy, slightly raised | pink-white-yellow | maroon |
| | S5, S10, S20 | compact; elevated | pink-white-yellow | maroon |
| | W5, W10, W20 | fluffy, elevated and compact | pink-white-yellow; white-yellow | maroon |

C5, C10, C20—control with 5%, 10%, 20% solvent residue; Y5, Y10, Y20—5%, 10%, 20% yarrow extract; T5, T10, T20—5%, 10%, 20% tansy extract; S5, S10, S20—5%, 10%, 20% sage extract; W5, W10, W20—5%, 10%, 20% wormwood extract.

## 4. Discussion

The conducted laboratory experiments allowed for determining the direct effect of plant extracts on the growth dynamics of fungi of the genus *Fusarium*, important in the pathology of agricultural plants. Studies of the fungistatic activity of plant extracts are conducted all over the world and concern various pathogens. The most important in this regard are herbal plants, which show strong biocidal properties due to the content of various biologically active compounds [36–43]. Currently, it is well recognized that hundreds of biological active chemical compounds are present in plants, working in synergism, and conferring a broad variety of bioactivities. Phenolic compounds are the most common group of plant components with strong antimicrobial, including antifungal properties. The present study has shown that the sage (*Salvia officinalis*) extract demonstrated best fungistatic properties. It is a medicinal plant with antioxidant, antimicrobial and anti-inflammatory properties. Its diverse biological activity is mainly conditioned by polyphenolic compounds [44]. The main components of the secondary metabolites of *Salvia* spp. are terpenoids and flavonoids. The aerial parts of these plants contain mainly flavonoids, triterpenoids and monoterpenes, especially concentrated in flowers and leaves [45–47]. Numerous studies also confirmed the high biocidal activity of sage secondary metabolites [46,47]. However, differences regarding this effect were identified depending on the dosage plant extracts applied. It was reported in different studies that biologically active compounds of *Salvia* spp. exerted antifungal effect on *Fusarium* species, such as *F. tricintum*, *F. sporotrichioides* and *F. oxysporum* f. sp. *radicis-lycopersici* [48,49]. Studies have also shown the antimycotoxigenic properties of major monoterpene constituent 1,8-cineole from *S. officinalis* in relation to ochratoxin A (OTA), produced by *Aspergillus carbonarius* [47]. Secondary metabolites of sage, especially phenolic acids, flavonoids and terpenes, also show high antioxidant activity [50–54]. Rowshan and Najafian [50] showed in their study the highest content of such polyphenols as rosemarinic acid, catechin, vanillin, chlorogenic acid, quercetin and p-coumaric acid. The antioxidant capacity of the tested extracts was correlated with the total phenolic and flavonoid content [55], and these reports were confirmed in the present study. Sage, due to the highest phenolic content among the tested plant extracts, was also the most effective against pathogenic fungi, inhibiting their growth even by 83.53% (*F. culmorum*—20% concentration).

High antifungal activity was also observed for tansy extracts, especially for the concentration of 20%. Tansy is characterized by a high content of secondary metabolites [56,57], thus it exerts a strong antibacterial and antifungal effect [58,59]. Tansy plants contain many phenolic compounds (flavonoids and phenolic acids). Among flavonoids, there are mainly luteolin, apigenin and quercetin glycosides, while phenolic acids are predominantly represented by chlorogenic, caffeic and dicaffeoylquinic acids [57,60]. *T. vulgare* extracts also show high antioxidant activity, strongly correlated with the content of polyphenols, especially flavonoids [61]. The high antioxidant potential (FRAP and DPPH) of tansy extract was shown by Bączek et al. [60], and the present study confirmed the aforementioned reports. However, the world literature lacks detailed data on the effect of tansy extracts on phytopathogens, especially fungi of the genus *Fusarium*. Korpinen et al. [56] showed a strong inhibitory effect of tansy extracts on *Penicillium venetum* and *Aspergillus niger*. However, Wens and Geuens [62] showed that the addition of 500 μL of *T. vulgare* extract

to the culture medium inhibited the growth of *F. oxysporum* but only at the level of 32% compared to control. Different results were obtained in the current study, indicating a strong antifungal effect of tansy extract, especially during the first days of the experiment (2nd day), mainly against *F. culmorum* (72.5%), *F. avenaceum*, (67.5%) and *F. sporotrichioides* (64.5%) at 20% concentration of the extract in the medium. Furthermore, it should be noted that one plant from a given genus has a specific chemical composition and its significant medicinal value does not mean that all the other plants of the same genus had the same properties. Some studies even reported that the same species, but of different origin showed considerable differences in chemical composition, and thus, different bioactive efficacy and potency [43,63].

Yarrow is also a rich source of flavonoids. *Achillea millefolium* is a medicinal plant containing over 100 different biologically active compounds with antimicrobial properties [64]. Flavonoids, apigenin and quercetin, phenolic acid and caffeoylquinic acid have been described as the main phenolic compounds present in yarrow [65]. The chemical composition of a plant can vary significantly due to various factors, such as the geographic origin of the plant, the time of harvest and the analyzed plant parts, extraction and analysis methods [43,66]. This was confirmed by Georgieva et al. [67], indicating that the total content of polyphenols in *A. millefolium* water extracts (2.74–7.92 mg GAE/100 g fw) and the total content of flavonoids (from 0.05% to 0.07%) changed over time. The highest content of these compounds was determined at the beginning and the lowest at the end of growth. The high content of flavonoids, polyphenols and the high antioxidant potential of water extracts in our study was determined at a level similar to that presented by other researchers [68,69]. Fierascu et al. [69] showed that yarrow extract strongly inhibited the growth of *Aspergillus niger* (70.19%) and *Penicillium hirsutum* (47.40%) [69], as well as the oil had fungistatic effect against *Rhizopus stolonifer* (65.7%) and *Verticillium dahliae* (65.3%) [63]. High content of secondary metabolites determines the strength of the extract's antifungal effect, which was also confirmed by our research. The strongest antifungal effect of the extract was recorded for a concentration of 20% at the beginning of the experiment, for all *Fusarium* species tested, especially against *F. sporotrichioides* (60.2%). The extracts showed the weakest effect against *F. graminearum*, 5% concentration of the extract in the culture medium even stimulated mycelial growth in subsequent days.

Of the plant extracts tested, woodworm extract showed the weakest antifungal properties against *Fusarium* spp. Kordali et al. [70] confirmed the fungistatic properties of wormwood from the Turkish population against 34 fungal species, including *F. solani* and *F. oxysporum*. Although many authors confirmed the fungistatic effects of wormwood extracts and oils [70–73], the strength of their biocidal action varied. The high efficacy of *A. absinthium* ethanol extracts has been described in laboratory studies against such phytopathogens as *Penicillium expansum, Botrytis cinerea, B. allii, Monilinia laxa, M. fructigena* or *Plasmopara viticola* [72]. The main antifungal compounds contained in wormwood extracts include flavonoids, thiophene and terpenoid [74,75]. Canadanovic-Brunet et al. [27] showed that the biocidal activity of extracts was elevated with increasing content of phenolic compounds and flavonoids, indicating that they contributed to antiradical and antioxidant activity.

Plant extracts from sage, yarrow, tansy and wormwood, in addition to inhibiting the growth of *Fusarium* spp. fungi, caused changes in the colour and structure of aerial mycelium. Jamiołkowska and Kowalski [31] reported that the addition of grapefruit extract (Biosept 33 SL) to the culture medium strongly inhibited the growth and formation of morphological elements of *F. avenaceum, F. equiseti, F. culmorum* and *B. cinerea*, thereby limiting fungal sporulation. The authors also found structural changes in the hyphae, such as cytoplasm dehydration and hyphae deformation. The inhibitory effect of biopreparations on the growth of phytopathogenic fungi in vitro was studied by Orlikowski et al. [76,77]. These authors showed that the compounds present in the grapefruit extract (Biosept 33 SL preparation) inhibited not only the germination of *F. oxysporum* and *B. cinerea* spores, but also the growth of germ hyphae by dehydrating the cytoplasm of mycelial cells.

The strength of the biocidal effect of plant extracts varies and depends on the type of extract and the microorganism it acts on. The results of the present study and literature data indicate that the chemical composition of plant extracts varies depending on the plant species and the sampling site [78]. While the antibacterial and antioxidant effects of various plant extracts are well understood, the antifungal and antimycotoxigenic have not yet been thoroughly studied. The present study complements the data on the bioprotective effect of plant extracts against selected phytopathogens of the genus *Fusarium*, important in agricultural phytopathology. However, the results obtained require confirmation of the antifungal effect of the extracts under field conditions, and a properly prepared mixture of plant extracts may increase the strength of fungistatic effect and provide the basis for the development of a new biofungicide of plant origin and become a safer alternative for sustainable and organic crop production.

## 5. Conclusions

The use of natural compounds for pathogen control is very attractive, and the availability of biotechnological methods open new avenues for plant protection approaches. The results of the conducted experiments indicated a positive effect of plant extracts from sage, tansy, yarrow and wormwood on the growth of fungi of the genus *Fusarium* spp. The extracts inhibited the growth of *Fusarium* spp. fungi significantly better at a higher concentration (20%) compared to lower concentrations. Sage and tansy extracts showed the strongest antifungal activity against *F. culmorum*, *F. avenaceum* and *F. sporotrichioides*, *F. graminearum*, and wormwood extracts exerted the weakest effect. It was also demonstrated that the greater fungistatic effect of plant extracts depended on the higher content of secondary metabolites (polyphenols and flavonoids) and their high antioxidant activity. The conducted research is the basis for further research to determine the fungistatic effect of plant extracts in field conditions, their phytotoxicity and biological stability. Plant extracts obtained in a conventional way may, however, be unstable and difficult to obtain on an industrial scale, therefore, efforts should be made to develop such biotechnological methods that ensure stable biologically active compounds, the production of which will be economically justified. The results of field research will allow for the development of the proper composition of the plant component (mixture of extracts), which will be the basis for the production of a natural preparation for cereal protection against fusariosis as well as seed treatment.

**Author Contributions:** Conceptualization, A.J. and W.K.; methodology, A.J., J.W. and R.K.; software, W.K., A.J. and J.W.; validation, A.J. and R.K.; formal analysis, W.K., J.W. and A.J.; investigation, W.K. and A.J.; resources, W.K. and A.J.; data curation, A.J. and J.W.; writing—original draft preparation, W.K. and A.J.; writing—review and editing, A.J. and W.K.; visualization, A.J.; supervision, A.J.; project administration, A.J.; funding acquisition, A.J., W.K. and R.K. All authors have read and agreed to the published version of the manuscript.

**Funding:** This research was supported by projects no.SD/38/RiO/2022, no OKK/s/44/2022 provided by University of Life Sciences in Lublin, Poland.

**Conflicts of Interest:** The authors declare no conflict of interest.

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
