# Peer review of "Antifungal Effect of Plant Extracts on the Growth of the Cereal Pathogen Fusarium spp.—An In Vitro Study"

_agronomy, doi:10.3390/agronomy12123204_

Round 1

Reviewer 1 Report

the title should mentioned that this is ethanol extract 

Abstract should be have conclusion 

authors should mentioned why they selected these plants 

point 2.2 what you mean by Mycological materials 

is this isolates identified before 

how you prepare the concentration ?

table and figs need more details 

please see attach file 

Author Response

Rewiev 1

Thank you very much for your detailed review. Valuable comments are very helpful in editing the article and will allow to prepare a high-quality article. All your comments have been included in the article.

Title: the title should mentioned that this is ethanol extract 

As the authors of the paper, we propose do not change the title because the tested extracts were water extracts, although 70% ethanol was used for their preparation. During the solvent evaporation procedure according to the methodology described in the article, there was no ethanol in the working extracts added to the microbiological medium.

Abstract: Abstract should be have conclusion 

The abstract has been revised, a brief summary has been included and a further research goal has been set.

Material and methods: authors should mentioned why they selected these plants 

The plants used for the study were selected on the basis of information of the literature and the valuable biochemical composition of the herb plants. In addition, these are plants that commonly grow in the natural state (sometime as weeds) of a temperate climate and are easy to collect herbal material. This explication is provided in the article.

Point 2.2 what you mean by Mycological materials

The term mycological materials has been changed to fungal cultures.

Is this isolates identified before?

The authors confirmed that the studied Fusarium strains belong to the correct species based on microscopic observations (spore shape and size) and appropriate mycological keys. This explanation was included in the article.

2.1.2. how you prepare the concentration ?

The concentrations of the extract were prepared in accordance with the appropriate methodology. Crushed plant material (300.0 g) of each type of herb was weighed into round-bottomed flasks and 3000 mL of 70% ethanol was added. Extraction was carried out under a reflux condenser at the boiling point of ethanol for 6 hours. The resulting extract was filtered through filter paper and concentrated to 300 mL (extract 1:1) using a rotary evaporator. In this way, a 100% extract was obtained. But the 100% extract was diluted and tested at concentrations of 5%, 10% and 20%.

Table and figs need more details 

Table 1-5: abbreviations SD and numerical values a,b,c… are explained below the tables.

Tt is also marked where the discussion section begins.

In the opinion of the authors, the use of Latin names of plants or common names interchangeably is allowed.

All comments and remarks in the text have been taken into account and corrections have been made in the article.

Reviewer 2 Report

Comments to the author Agronomy (2097832)

The manuscript (by Kursa et al) entitled “Antifungal effect of plant extracts on the growth of the cereal pathogen Fusarium spp. – an in vitro study” is really interesting work and the author (s) performed a nice in vitro study. The study by Kursa et al. demonstrates the study of various plant extract on the most important fungal pathogen. Overall, the manuscript is not well structured; presenting novelty and authenticity of work. The results are reliable and manuscript is in accordance with the Journal’s scope. However for the improvements, some shot coming in text are needed to be fixed before further consideration.

In abstract

Key words: Remove the names of extract and write appropriate words, Abstract words should not be the duplicates of title.

Suggested keywords: Phyto extracts, Fusarium, Antifungal activity  

Line 21: In first case, write down the complete form of DPPH, later on use abbreviation.

Line 25: Inhibited the mycelia growth of fungal pathogen depending upon the specie.

Line 26-27: The extracts of sage and tansy plants demonstrated strong inhibitory effect at higher (20%) concentration followed by lower concentrations (5% and 10%).

(Here, the authors have to mention the numerical data of strong inhibition to justify and compare with other concentrations).
The abstract is too long, it must be concise and to the point in terms of results. However, this needs to be short and justified with numerical data of results. (Only write the best treatment compared with control)

Line 28: “Based on the study, it was found that”à the results of present study demonstrate the significant variation in the mycelial growth of fungal species after the addition of various phyto extracts.

No conclusion/recommendations in abstract?

Need to mention the objectives point by point

Line 95-97: The results ?

Material and methods

2.1.1: need to Mention which part of plant was used as extract as different parts ahs different antifungal potential

The structure and sequence should be as: Isolation of fungi, collection of plants for extract preparation; analysis of antioxidants in extract………..

Section 2.2.1 How the author claim the confirmation of species?
as Fusarium contains hundreds of species how morphologically confers the exact identification?

If the author obtained previously used/identified cultures, then provide reference.

Line 139-142: Unclear statement.!

Section 2.2.2: Biotic activity of plant extracts in vitroà In vitro evaluation of antifungal potential of plant extracts

The author used 5%, 10% and 20% but why not 100%? (What if the author use 100% of extract)

Line 151-152: The control should be extract free/only on PDA without any amendment of plant extract but here is different why?

Figure 1: The author used the formula of inhibition (Line 160), but in figure 1, they mentioned the colony diameter?  

Figure 1: Need to add original figure, readers are unable to see the trend of results even after zoom.

Line 223: fungicidal activity?

Need to change the inhibition tables into figures to clarify the inhibitions by plant extract

Line 334: “The current study also confirmed reports of other”?

Conclusion:

The repeated results-not conclusions

Author Response

Review 2.

Thank you very much for your detailed review. Valuable comments are very helpful in editing the article and will allow to prepare a high-quality article.

Abstract

Key words: Remove the names of extract and write appropriate words, Abstract words should not be the duplicates of title.

Keywords: have been changed as suggested by the reviewer

Line 21: the abbreviation DPPH was explained in the article and its full name was given

Line 25: corrections were made in accordance with reviewer's comments

Line 26-27: corrections were made in accordance with the reviewer's comments;  the most important data on the potency of inhibiting fungal growth was given in the abstract.

Line 28: corrections were made in accordance with reviewer's comments.

No conclusion/recommendations in abstract?

Line 29: in the abstract, a brief summary is provided and the further aim of the research is set.

Line 95-97: the abstract was supplemented with results on the inhibitory effect of extracts on fungal growth.

Material and methods

Section 2.1.1. need to mention which part of plant was used as extract as different parts ahs different antifungal potential

In this subchapter explained what part of the plant material was used for research (leaves).

The structure and sequence should be as: Isolation of fungi, collection of plants for extract preparation; analysis of antioxidants in extract………..

The reviewer proposed to change the order of the subchapter in this part of the research, but according to the authors, this order is correct and will not change the value of the article.

Section 2.2.1. How the author claim the confirmation of species?

The authors explained that the tested strains of fungi were designated as proper Fusarium species on the basis of microscopic observations (spore shape and size) and appropriate mycological keys. This explanation was included in the article.

Line 139-142: This part of the article explains source origin of wheat grains (Breeding Companies).

Section 2.2.2. the subsection title has been changed in accordance with the reviewer's comments

The author used 5%, 10% and 20% but why not 100%? (What if the author use 100% of extract)

The experiment used 5%, 10% and 20% extract, not 100% extract. The most effective extracts in the appropriate concentration will be tested in field trials. Concentrations were determined based on literature information. It is also assumed that too high concentration of the extract (100%) would cause a phytotoxic effect.

Line 151-152: The control should be extract free/only on PDA without any amendment of plant extract but here is different why?

The artificial medium (PDA) of the control contained a solvent residue previously concentrated in the same manner as in the preparation of the extracts, therefore no ethanol was present in the medium. This explication was added in the article in the methodological part.

Figure 1: The author used the formula of inhibition (Line 160), but in figure 1, they mentioned the colony diameter? 

The Figure 1 shows the diameters of the fungal colonies (fungal growth dynamics), NOT the percent inhibition of mycelial growth calculated according to Abbott's formula. Percentage of mycelial growth inhibition are presented in tables 2-5. Figure1 have been changed to be more readable.

Line 223: Term Fungicidal activity has been changed to fungistatic activity

Need to change the inhibition tables into figures to clarify the inhibitions by plant extract

Tables 2-5 present the percentage of inhibition of fungal growth colonies compare to the control. After discussion with others authors of the article, we would like to  present the results in the form of tables, not graphs, because in our previous studies of this type, the results were presented in the form of tables, therefore they will be easier to comparisons.

Line 334: Sentence changed as suggested by reviewer.

Conclusions: The repeated results-not conclusions

Conclusions have been revised in accordance with the reviewer's comments

Round 2

Reviewer 2 Report

Comments to the author Agronomy (2097832)

The manuscript (by Kursa et al) entitled “Antifungal effect of plant extracts on the growth of the cereal pathogen Fusarium spp. – an in vitro study” has been significantly improved but still need a proof reading for some minor improvements.

In introduction, no information is found regarding the use of plant extract (used in this study) is this first time to use these extracts or previously used. There should be information of these extracts (if previously used) to validate their antimicrobial potential.

Line 25: Mycelial growth

Line 26-27: remove “and duration of action”

Line 113: collected prior to flowering of plant/Plants.

Line 114: Ground à Grounded in what? Liquid nitrogen or in some solution?

Line 119: Crushed plant materialà powdered material; 300.0 à 300 g

Line 119: To prepare the extracts, 300g of each powdered herb was suspended in 3000ml of 70% ethanol.

Line 123: Heidolph Companyà removed the word company and add the country etc along with the name of company

Line 319-320: The present study focused on the use of “Plant extract”, the reference/discussion on essential oils is redundant because essential oils is a totally separate treatment than plant extracts which contain various compounds. 

In all figures simply write down the numbers like 10, 20, 40 instead of 20.00, 30.00 etc.

More, in vertical side, do not add major tick it does not seems appropriate, only with horizontal side.

In the manuscript, whenever the author use the word “Significantly”, add (p ≤ 0.05).

I am still on my point why not used 100% as at 20% amazing results were obtained but what if the author used 100% concentration. More how is it possible that plant extract at 100% will cause toxicity to plants, although, plant have antifungal potential against the diverse genus Fusarium.

I would suggest to work with 100% too in future because biocontrol approaches will provide a sustainable approach to control fungal plant pathogens specially the Fusarium.

Author Response

Review 2.

In introduction, no information is found regarding the use of plant extract (used in this study) is this first time to use these extracts or previously used. There should be information of these extracts (if previously used) to validate their antimicrobial potential.

Research on plant extracts and their biocidal effect has been conducted for many years. Currently, scientists are focusing on finding out the type of secondary metabolites that are responsible for the biocidal activity of the plant extracts. The purpose of the introduction to the article was not to duplicate information on the biocidal effect of plant extracts, but rather to indicate the richness of secondary metabolites contained in many plants, including herbal plants, and their biocidal effect. They can be produced in bioreactors in order to obtain stable and effective biological preparations. Therefore, in the research I present not only the fungistatic effect of plant extracts from sage, tansy, wormwood or yarrow, but also the content of the most important secondary metabotites such as polyphenols. More information on the antifungal activity of the tested plants is included in the discussion.

Line 25: Mycelial growth

Corrections have been included in the text

Line 26-27: remove “and duration of action”

Corrections have been included in the text

Line 113: collected prior to flowering of plant/Plants.

Corrections have been included in the text

Line 114: Ground à Grounded in what? Liquid nitrogen or in some solution?

Word grounded changed to powdered; the plant material was mechanically powdered

Line 119: Crushed plant materialà powdered material; 300.0 à 300 g

Corrections have been included in the text

Line 119: To prepare the extracts, 300g of each powdered herb was suspended in 3000ml of 70% ethanol.

Corrections have been included in the text

Line 123: Heidolph Companyà removed the word company and add the country etc along with the name of company

Company names have been corrected and supplemented

Line 319-320: The present study focused on the use of “Plant extract”, the reference/discussion on essential oils is redundant because essential oils is a totally separate treatment than plant extracts which contain various compounds. 

Corrections have been included in the text

In all figures simply write down the numbers like 10, 20, 40 instead of 20.00, 30.00 etc.

I do not understand the reviewer's comment; when it comes to the reviewer about the numbering of the vertical axes in the figures, they are set automatically and I cannot change them.

More, in vertical side, do not add major tick it does not seems appropriate, only with horizontal side.

I do not understand the reviewer's comment

In the manuscript, whenever the author use the word “Significantly”, add (p ≤ 0.05).

Corrections have been included in the text

I am still on my point why not used 100% as at 20% amazing results were obtained but what if the author used 100% concentration. More how is it possible that plant extract at 100% will cause toxicity to plants, although, plant have antifungal potential against the diverse genus Fusarium.

I would suggest to work with 100% too in future because biocontrol approaches will provide a sustainable approach to control fungal plant pathogens specially the Fusarium.

100% extract cannot be used on plant tissues because, as already explained, it causes a strong phytotoxic effect! The concentration of secondary metabolites in living plants is not as high as in plant extracts, so the reviewer's comment that it harms plants is inappropriate. Of course, many plants have a toxic effect on pathogens, but not enough, so we support the plant with a traitment containing biologically active compounds with fungicidal/fungistatic properties. If the concentration of a toxic substance in living plants is high (in natural conditions), it indicates the effect of the plant's immunity against a specific pathogen.
